# A Safe Return to Campus in Times of COVID-19: A Survey Study among University Personnel to Inform Decision Makers

**DOI:** 10.3390/vaccines10030371

**Published:** 2022-02-27

**Authors:** Tugce Varol, Francine Schneider, Ilse Mesters, Robert A. C. Ruiter, Gerjo Kok, Gill A. Ten Hoor

**Affiliations:** 1Department of Work and Social Psychology, Maastricht University, P.O. Box 616, 6200 MD Maastricht, The Netherlands; t.varol@maastrichtuniversity.nl (T.V.); r.ruiter@maastrichtuniversity.nl (R.A.C.R.); g.kok@maastrichtuniversity.nl (G.K.); 2Department of Health Promotion, CAPHRI, Maastricht University, P.O. Box 616, 6200 MD Maastricht, The Netherlands; francine.schneider@maastrichtuniversity.nl; 3Department of Epidemiology, CAPHRI, Maastricht University, P.O. Box 616, 6200 MD Maastricht, The Netherlands; ilse.mesters@maastrichtuniversity.nl

**Keywords:** COVID-19, university personnel, beliefs, safety, vaccination, return to work

## Abstract

Halfway through 2021 in the midst of a public health crisis, a new academic year was fast approaching. Dutch universities were preparing to reopen their campuses to students and personnel in a safe manner. As the vaccination uptake was increasing and societies were slowly reopening, inviting students and personnel to campus became the next step to “the new normal”. To absorb this change seamlessly, it was considered important to investigate personnel’s beliefs about returning to campus and their perceptions of a safe working environment. An online survey was conducted among personnel (*N* = 1965) of Maastricht University, the Netherlands. University personnel’s beliefs about a safe return to campus were assessed. The data were collected between 11 June and 28 June 2021. This study showed that, while most personnel (94.7%) were already vaccinated or willing to do so, not all personnel did feel safe to return to campus in September 2021. Over half of the respondents (58%) thought that the university is a safe place to return to work when the new academic year starts. However, the remainder of personnel felt unsafe or were uncertain for various reasons such as meeting in large groups or becoming infected. Moreover, when returning to campus, employees stated that they would require some time to reacclimate to their former work culture. The group who felt relatively more unsafe indicated that returning in September was too risky and that they worried about being infected. They wanted the safety guidelines to still be in force. On the other hand, the “safe” group stated safely returning to be “certainly possible” and trusted that others would still stick to the prevention guidelines. The findings led to practical recommendations for the University Board as they were preparing for organizing research and teaching for the upcoming academic year in the context of the COVID-19 pandemic. A brief intervention was developed: a webinar in which the data were linked to the board’s plans for safe returning. This study demonstrates that university boards may use research among personnel to develop adequate measures promoting safety and feelings of safety among personnel in similar future situations.

## 1. Introduction

Since the start of the COVID-19 pandemic, governments have been implementing mitigation rules to curb the number of COVID-19 cases and hospitalizations. Among those rules, the closure of higher educational institutions was implemented worldwide, which led many universities to switch to online education in order to prevent disruption in students’ learning [1,2]. In the Netherlands, Dutch universities, including Maastricht University (UM), chose to offer hybrid education at the start of the academic year 2020–2021. However, in November 2020, with a steep rise in the number of COVID-19 cases, all universities had to move their education to fully online. At this time (November 2020), stage-wise COVID-19 vaccination was offered to everyone living in the Netherlands. The Dutch government aimed at vaccinating everyone over the age of 18 who was willing to get vaccinated against COVID-19 by September 2021 [3].

As the new academic year (1 September 2021–31 August 2022) was approaching, Dutch universities were eager to welcome students back to the campus after summer, if allowed by the government, in light of the increase in the COVID-19 vaccination uptake (in the week of 11 May 2021, 84% of all people over the age of 16 were vaccinated or willing to vaccinate soon) and the fall in the number of positive COVID-19 cases and hospitalizations in May 2021 [4]. Brammer and Clark [5] shared their reflections concerning the impact of the COVID-19 pandemic on students and university personnel. They stated that the uncertainties, concerns, and increased workload posed by COVID-19 caused stress and anxiety among students and personnel. Due to the adverse effect of the COVID-19 pandemic on the education and wellbeing of students and staff [6,7,8], universities were looking for secure methods to welcome students and university personnel back to their campuses in September 2021.

Problem solving and policy development require a thorough understanding of the problem. Intervention mapping, a framework for theory- and evidence-based program development at different environmental levels, embraces the involvement of stakeholders in the problem diagnosis and planning for the solution [9,10,11]. As higher education institutes are not only home for students but also for university personnel, their perspectives were deemed important by the Board of UM in order to facilitate a smooth transition to work on-site, create a safe environment, and optimize vaccination decision making.

In this exploratory study among UM personnel, the aim was to collect data to assist the university executive board’s policy/decision making. For this, we explored the feelings of safety of university personnel when trying to imagine returning to UM in September by asking them about (a) positive and negative attitude beliefs, (b) trust and worries, and (c) preventive measures, especially COVID-19 vaccination uptake. The findings of this study helped the University Board to respond to the upcoming reopening of the university, and the study procedures may be reused for comparable pandemic and epidemic threats in the future.

## 2. Methods

### 2.1. Procedure and Participants

Personnel (n = 7.198) of the university including both academic personnel and support staff, such as policy managers, secretaries, and IT experts working at the university, were invited to participate in the study through two channels: (1) an existing employee panel of the university operated by a certified survey agency (Flycatcher; https://www.flycatcher.eu/en/Home/OverOns, accessed on 27 January 2022), and (2) an email that was sent on behalf of the executive board to all personnel.

An online survey was used to collect information, which began on 11 June and ended on 28 June 2021. One reminder was sent out on 21 June. First, all panel members were emailed a unique hyperlink. Subsequently, a general hyperlink was emailed to all personnel; personnel who were members of the panel were instructed to use the URL provided by the survey agency. Participants agreed to participate in the study by clicking on the hyperlink included in the invitation and the agreement box before they could begin the questionnaire. This study was approved by the Ethics Review Committee, Psychology and Neuroscience, Maastricht University, ERCPN: 188_10_02_2018_S68.

### 2.2. Measurements

The focus of this cross-sectional study was to explore (1) whether university personnel would feel safe when imagining returning to campus in the new academic year (2021) and (2) the relevant safety beliefs (or exploratory constructs) split into (a) positive and negative attitude beliefs, (b) trust and worries, and (c) thoughts on preventive measures including COVID-19 vaccination uptake.

***Preparation.*** The questionnaire was developed based on the available literature, theory, and the information gathered through interviews among university personnel [12]. In the preparation of this study, our search for literature on university personnel’s sense of safety upon returning to work yielded no results. However, there was literature on COVID-19 vaccination or vaccination intention in general [13,14,15,16,17], as well as on people’s responses to fear appeals [18]. Further, the construction of the questionnaire was guided by social cognitive theories [19,20,21,22] as well as the online interviews that were conducted with UM personnel (*N* = 8; unpublished data). In the interviews, personnel were asked their opinions about the safety of the work environment when returning to campus in September 2021, as well as about the COVID-19 vaccines.

The questionnaire was reviewed by several experts and revised based on the feedback received. Both English and Dutch versions of the questionnaire were available for personnel to fill out. The Supplementary Materials contains the complete questionnaire, File 1. The questionnaire, Supplementary Materials, and non-identifiable data can be found at the Open Science Framework: https://osf.io/hb5rw/ (accessed on 27 January 2022). The Strengthening the Reporting of Observational Studies in Epidemiology (STROBE) guidelines for observational studies were used while reporting this study [23].

***Returning to campus in September.*** Participants were asked, “When I try to imagine the situation in September, I think that UM is a safe place to return to”. A 7-point Likert scale (fully disagree (1)–fully agree (7)) was provided as the answer option.

***Positive and negative attitude beliefs*** were measured with ten belief questions. Example items are: “Starting again in September full-on… is too fast/requires a transition period/is too risky/means that I have to protect myself against others”; “Returning to “normal in September” is certainly possible”; and “I am happy that I can see my colleagues in real life again”. All questions were responded to on 7-point Likert scales (fully disagree (1)–fully agree (7)), and for certain questions, a “not applicable” option was included.

***Trust and worries*** comprised eight belief questions, for instance, “I trust that UM will be a safe place in terms of people sticking to the prevention rules” and “I am worried about students and staff returning from high-risk countries” with the answer option (Likert scale): fully disagree (1)–fully agree (7). For some questions, a “not applicable” answer option was included.

***Preventive measures*** entailed questions regarding facilities (2 items), entrance testing proof (2 items), safety guidelines (2 items), and people with health complaints (3 items). A 7-point Likert scale (fully disagree (1)–fully agree (7)) was used as the answer option. One exception in terms of response option was for “*entrance testing*” items: “*I think that asking people to show entrance testing proof, or to do a test, is*” (1): *not feasible at all* (1)—*very feasible* (7); and (2): *not useful at all* (1)—*very useful* (7). (Note: for entrance to restaurants, events, or other activities/buildings, a mandatory test was suggested, where people had to show that they are vaccinated, recovered from a recent COVID-19 infection, or tested negative for the coronavirus).

***COVID-19 vaccination*** intention and/or behavior was measured with the item “*I have been vaccinated against COVID-19*”. The response options were (1) *yes, fully*; (2) *yes, partially;* and (3) *no*. Participants who chose “*no*” continued with the question “*You indicated that you have not (yet) been vaccinated. Which of the following statements is most applicable to you?*”, with four response options: (1) *“**I intend to take the vaccine when it is my turn”;* (2) *“**I have not been vaccinated and decided to not take the vaccine when it was my turn”;* (3) *“I do not intend to take the vaccine*”; and lastly (4) *“I do not know yet whether I want to get vaccinated”*. Vaccination beliefs were assessed by including 18 items, with a 7-point Likert scale (fully disagree (1)—fully agree(7)) response option.

***Demographics*** included age, gender, how long they have been employed by the university, whether they work full-time or part-time, their function at the university (*“teaching and research”*; *“academic support, policy and management”*; and *“other”(not further specified)*), where they work (*“a faculty”*; *“a service center”*; and *“other”*), and whether they see themselves as a member of a high-risk group for COVID-19 (*“yes”*; *“no”*; and *“I do not know”*).

### 2.3. Data Analysis

For all items, descriptive analysis was conducted to calculate the means (*M*), standard deviations (*SD*), and frequencies by using IBM SPSS Statistics 26. There were no missing data. In the preliminary analyses, we found that the members of the Flycatcher panel and the personnel group had quite similar outcomes. Given the comparable results, we did not differentiate between the Flycatcher panel members versus the other UM personnel in the data analysis.

Correlations between the question “*When I try to imagine the situation in September, I think that UM is a safe place to return to*” and all potential underlying beliefs were calculated for positive and negative attitude beliefs, trust and worries, and preventive measures. Additionally, we performed ANOVA with the Welch statistic (with post hoc Games–Howell test) to characterize three groups: unsafe, neutral, and safe.

Vaccination behavior/intention was grouped into three categories (yes, no, and do not know): “*Yes, fully*”, “*Yes, partially*”, and “*I intend to take the vaccine when it is my turn*” was grouped as “yes” (*N* = 1860); “*I have not been vaccinated and decided to not take the vaccine when it was my turn*”, and “*I do not intend to take the vaccine*” were grouped as “no” (*N* = 39); and “*I do not know yet whether I want to get vaccinated*” was grouped as “do not know” (*N* = 66). For vaccination beliefs, in order to compare the mean scores of the “yes”, “no”, and “do not know” groups, we started off by running ANOVA with the Welch statistic [24]. Subsequently, to detect which means differ from one another, we proceeded with a post hoc (Games–Howell) test. In order to examine whether the vaccination was also a factor in people’s beliefs about returning to work safely in September, we compared the results of the returning to campus questions with the results of the vaccination questions by using crosstab analysis.

## 3. Results

### 3.1. Demographics

Of the 7198 invited people, a total of 1965 personnel (27.3% response rate; 62.2% female) completed the survey; 21.5% of participants were in the age group of 56–65 or older than 65, and 14.4% identified themselves as a member of a high-risk group. Full background characteristics of the sample can be found in Table 1. Demographic characteristics of the participants of the survey and the total UM population were highly comparable.

### 3.2. Beliefs of University Personnel about a Safe Return to Campus

Of all personnel, 58.3% (score 5–7) indicated that Maastricht University is a safe place to return to work in September, while 23.9% (score 1–3) found the university not safe to return to in September 2021 (17.8% was undecided—score 4). The mean score was 4.63 (1–7) with a standard deviation of 1.64.

### 3.3. Negative and Positive Attitude Beliefs about Returning to Campus in September

Of the 1965 participants, 32.4% found starting again in September full-on too fast, and 28.2% indicated that it is too risky, while 43.2% indicated that returning to normal in September is certainly possible. Moreover, 48.3% stated that they have to protect themselves against others when they start working full-on in September, 65.1% thought that in order to start again in September full-on they require a transition period, and 41.5% were afraid that there will be too many adjustments for them when returning to campus. However, most of the participants stated that they can deal with being back in the office again. They were happy that they can start working at the office again (65.8%), that they can see their colleagues again (85.1%), and will have contact with students in real life again (74.9%).

All items were significantly correlated with *“When I try to imagine the situation in September, I think that UM is a safe place to return to”.* For all attitude beliefs, the “feeling unsafe” group is significantly more negative than the neutral group, while the “feeling safe” group is significantly more positive than the neutral group. The largest correlations with starting again in September are: “too risky”, (*r* = −0.73), as negative belief, and “is certainly possible”, (*r* = 0.66), as positive belief (See Table 2).

### 3.4. Trust and Worries about Returning to Campus in September

Personnel stated that the university will be a safe place in terms of people sticking to the prevention rules (61.9%) and facilities such as ventilation and disinfectants (73.7%). The main worry of the personnel was about meeting in large groups (63.3%), followed by students and staff returning from high-risk countries (61.8%) and being worried about how to deal with vaccine deniers/refusers (58.5%). Half of the participants indicated they are worried about becoming infected by COVID-19. *M*s and *SD*s can be found in Table 2.

For all beliefs on trust and worries, the feeling “unsafe” group is significantly more worried and less trusting than the neutral group, while the feeling “safe” group is significantly more trusting and less worried than the neutral group. The largest correlations with starting again in September are: “people sticking to the prevention rules” (*r* = 0.65), as trust belief, and “I still worry about being infected”, (*r* = −0.62), as worry belief (See Table 2).

### 3.5. Preventive Measures Related to Returning to Campus in September

The ventilation at work was perceived to be good enough to prevent becoming infected by 26.7% of personnel, while half of the participants indicated that the rules about ventilation in the buildings are not clear. Asking people to show entrance testing proof, or to do a test, was found to be not feasible by 54.9%, but to be useful by 46.8% of participants. In terms of the rules that should still be implemented in September, distancing (64%) and facemask (49.4%) rules were viewed as necessary by (more than) half of the personnel. In case people (personnel and students) have symptoms such as sniffling or coughing, 84.5% of personnel stated that these people should stay at home and should not come to campus, and if people are sniffling or coughing on campus, 74.6% stated that they should be sent home. Almost all participants (90.6%) indicated that the university should provide clear guidelines about how to deal with students who have health complaints.

The differences between the “unsafe”, “neutral” and “safe” groups for all beliefs can be found in Table 2. The largest correlations with starting again in September are the statements about safety guidelines, “still keeping 1.5m distance” (*r* = −0.52), and “still wearing a facemask” (*r* = −0.50). Surprising were the results of asking for entrance testing proof: “not useful” (*r* = −0.16), and “not feasible” (*r* = −0.01), meaning that, at that time, personnel’s views on entrance testing proof were not related to their feelings of safety on returning to work.

### 3.6. COVID-19 Vaccination Uptake and Beliefs

Of 1965 personnel, 1860 (94.7%) indicated that they are either already vaccinated against COVID-19, or they are willing to do so. Only 2% had decided not to take the vaccine when it was their turn or did not intend to get the vaccine (and 3.4% were undecided). The vaccination beliefs of university personnel are depicted in Table 3. The mean scores of COVID-19 vaccination beliefs for each of the three groups (“yes”, “no”, “do not know”) differed significantly. Most personnel thought that being vaccinated against COVID-19 is the only way out of this pandemic (85.4%) and vaccination gives a feeling of safety (86.2%). Personnel (60.2%), also including those who already received a COVID-19 vaccine or intend to do so, did not agree with the statement that being vaccinated against COVID-19 does make it 100% safe. Moreover, most personnel (88.2%) thought that being vaccinated against COVID-19 would result in people keeping less distance from others.

Personnel who already received a COVID-19 vaccine or did intend to do so (“yes” group: *M* = 2.75; *SD* = 1.66) were not as worried about the safety of the COVID-19 vaccine as people who decided not to get vaccinated against COVID-19 or who did not intend to do so (“no” group: *M* = 6.51; *SD* = 1.02) and people who were undecided to get vaccinated (“do not know” group: *M* = 5.83; *SD* = 1.16). Likewise, both the “do not know” and “no” groups were more worried about the possible long-term negative side effects of COVID-19 vaccines as opposed to people who already received a COVID-19 vaccine or intended to do so. In terms of perceived norms, the “yes” group indicated that most people like them will get a COVID-19 vaccination (*M* = 6.20; *SD* = 1.02) and that most people who are important to them, want them to get a COVID-19 vaccination (*M* = 5.95; *SD* = 1.43). Moreover, contrary to the “yes” group, both the “no” and “do not know” groups did not agree that getting a COVID-19 vaccine is their moral duty (see Table 3).

We compared the results of the returning to campus questions with the results of the vaccination questions to inspect whether vaccination was also a factor in people’s beliefs about returning to work safely in September and found no relation between those; the vaccination percentages were uniformly high among all three returning to campus groups.

## 4. Discussion

Reopening universities safely in times of COVID-19 is a complex process and requires not only infrastructure changes but also consideration of stakeholders’ perspectives during the decision-making process. The findings of this study point towards not only focusing on real risks but also on “psychological” feelings of risk of university personnel. In this study, we explored university personnel’s views and worries pertaining to returning to campus in the new academic year (Fall 2021) and their thoughts on COVID-19 vaccination. Although more than half of employees indicated that the university is a safe place to work in September, the findings of this study revealed that a substantial number of personnel considered the university building unsafe or were uncertain about how safe it would be to start again. We also found that 95% of personnel that participated in the survey were vaccinated or were going to get vaccinated. To our knowledge, there are no comparable studies published yet in this (or similar) setting and/or context.

Although more than half of personnel indicated that starting to work full-on on campus in the new academic year is neither too soon nor unsafe, a large minority of personnel stated that they have to protect themselves against others while working on campus; which was in line with their worries about getting infected by SARS-CoV-2, despite the fact that COVID-19 vaccines were available and accessible to university staff and students at the time. Moreover, in addition to the infrastructure and COVID-19 regulations within the university, the main worries of personnel concerned meeting in large groups, exposure to students and staff who are returning from high-risk countries, and how to deal with vaccine deniers/refusers. They stated that they require clear guidelines from the university about how to deal with students who have health complaints.

In the Netherlands, all university personnel and students can get vaccinated against COVID-19. In this study, we found that most university personnel were either already vaccinated or intended to get vaccinated (94.7%). In our earlier study of university students [25], 80% of students indicated they would be willing to get the COVID-19 vaccine. Even though vaccination uptake did not show to be the major concern in this study, the UM board can still facilitate informed decision making around COVID-19 vaccination by targeting beliefs underlying vaccine hesitancy (e.g., side effects, the safety of the vaccines; see, for instance [25,26,27]. Moreover, due to the fact that people who are vaccinated can still be infected and spread the virus to others, it is advisable to implement COVID-19 regulations such as distancing, face coverings, testing, and isolating when offering on-site education as university personnel also viewed these measures as necessary (although only about half of the personnel viewed face masks necessary). Abandoning all COVID-19 regulations within the university when offering in-person education might increase the risk of infection at this stage of the pandemic when not all, or most, students are fully vaccinated in September 2021 as evidence shows that the highest effectiveness of the vaccines against the Delta variant (relevant for two-dose vaccines) is reached weeks after the uptake of two doses [28]. Furthermore, this might create anxiety among personnel and students who do not feel safe and are worried about being infected by SARS-CoV-2. Therefore, a stage-wise relaxation in the measures depending on the pandemic severity seemed advisable in educational institutions at that time.

The COVID-19 pandemic has demonstrated the importance of behavior change in combating the pandemic and having behavior change expertise in the planning group while developing and implementing theory- and evidence-based interventions [29,30,31,32,33]. Behavior change requires an understanding of the reasons behind people’s behavior and the psychological mechanisms through which behavior change can be reached by means of education and communication programs [34]. Thus far, several studies were conducted to identify the determinants of people’s compliance with preventive measures (e.g., [35,36,37]). The available empirical findings should be utilized with the guidance of behavior-change experts while planning the interventions [12]. The findings of the current study can be used by universities to provide their personnel with clear communication and guidance with regard to COVID-19 regulations, what to do when having symptoms, and how to deal with students who have health complaints indicative of COVID-19.

The COVID-19 pandemic has also led to a change in the work culture. Most personnel started working from home either fully or partly for more than a year and created a work habit and environment that best suits them. During the COVID-19 pandemic, university personnel and students experienced high levels of psychological distress [38]. As found in our study, university personnel require a transition period when returning to on-site work to get accustomed to their old work environment and habits. As we found that personnel have worries about working on campus in times of COVID-19, a prompt switch to on-site work might exacerbate their anxiety. We, therefore, did suggest that the UM board consider allowing personnel to temporarily work from home when not feeling safe yet, giving them the opportunity to get accustomed to “the new normal”.

Summarizing the results on returning to campus: employees from Maastricht University were willing (“happy”) to start working again on campus and see their colleagues and students in September 2021. However, they also saw risks and dangers, expressed in various descriptions of unsafe and unpredictable encounters and settings. Therefore, our policy recommendation to the board of the university was: give personnel an opportunity to reacquaint themselves with working in close quarters—start with a transition period in September and allow them to acquire work-on-site experiences.

### 4.1. Translating the Findings into a Brief Intervention

Based on the findings of this study, the UM’s marketing and communication department developed the following brief intervention for UM personnel and students to inform them about the measures and facilities that the university will provide to ensure a safe environment for the university’s anticipated September opening. The intervention consisted of a Webinar, on 6 July 2021, in which the results were summarized and presented by one of the researchers: UM-employees see risks and dangers, expressed in various descriptions of unsafe and unpredictable encounters and settings. The data show two explanations: (1) factual/epidemiological/medical reasons: uncertainties about the effect of vaccination in relation to new variants, and (2) psychological reasons: people have learned for more than a year to see others as a threat. That feeling cannot be switched off by a cognitive decision; people need some time to get accustomed again to social contacts. The policy recommendation was: if UM opens in September, give personnel a chance to reacquaint themselves to working close to others, i.e., start with a transition period during which people are not required to be present full time, but are instead encouraged to acquire work-on-site experiences in order to encourage them to return to work full time later. This was followed by a response from the Rector Magnificus of the university, explaining the measures that the university planned to take to provide a safe environment for personnel and students if the university could reopen after the summer break: https://www.youtube.com/watch?v=6OHCM7xXV1Q (accessed on 27 January 2022). University personnel were also referred to the vaccination webpage of the university which involves a frequently asked question section, videos developed with the involvement of experts from Maastricht University [32], and other informational resources on COVID-19 vaccines.

Immediately after the decision, on 13 August, by the Dutch Government that the universities were allowed to reopen in September 2021, personnel (and students) were informed about the measures taken via the university website and through email.

Based on the results of this study, personnel were told that if they had any concerns about safety despite all the precautions taken, they could contact their manager. Managers were provided with a guideline on how they can talk to personnel about these concerns and what they can do together so that people can return to work with peace of mind. If there are any medical reasons why personnel cannot come to work (or if they have symptoms like sniffing or coughing), they can make an appointment with the company doctor after having consulted their manager. Following up on the results of the “trust and worry” outcomes, a step-wise guide for (teaching) staff concerning how to deal with students who have symptoms was provided: step (1) teachers were advised to ask the student to leave the classroom and get tested; step (2) in case the teacher encounters a protest from the student, appropriate verbal responses to the student were provided with examples; and step (3) if the student still refuses to leave the building, the teacher was advised to contact the building manager who has the authority to order the student to leave.

In the end, there were some discrepancies between the advice of the researchers and the final decisions by the UM Board. This is not uncommon as governing bodies have to take into account other issues than safety as well. The advice of the researchers to the board was, when the situation was deemed to be safe, to reopen the university and give personnel who were hesitant some time to re-acclimate to work in a social setting. The board decided to open up the university on September 1st, as mandated by the government (despite the negative advice of the Outbreak Management Team; the governmental advisory board of experts), and to delegate the final decision about hesitant personnel to the company doctor, implying that only medical reasons were acceptable. Moreover, the formal (national) regulation for personnel not directly involved in teaching was to work from home as much as possible, which was formalized at Maastricht University as: 3 days at the office and 2 days from home.

### 4.2. Limitations and Strengths of the Study

This study has several limitations. We developed the questionnaire based on theories, empirical findings, and a limited number of interviews with university personnel. Although we interviewed personnel with different characteristics (e.g., cross-border workers, parents, different age groups, etc.), we might have missed some other viewpoints and worries of university personnel about returning to campus in the new academic year. However, we included an open-ended question at the end of the survey asking for any further remarks. Most of those remarks were about the positive aspects of working from home; others were about medical reasons for not vaccinating and the problems connected to providing informal care for family members. Those last two issues were taken up by the occupational health department. Second, we are in an insecure period due to uncertainties around new variants and thus, as a result, changes in the mitigation rules, staff members’ perspectives, and concerns may shift over time. Hence, university boards should monitor their personnel’s views toward working on campus in the future and adjust their strategies and policies accordingly. Third, this study was conducted in the Netherlands. Although we believe that the findings of our study would assist university boards in other countries as well while developing policies in their educational institutions, the feelings of safety of university personnel and their worries might vary depending on the COVID-19 risk level of the country and the vaccination level. Therefore, we suggest that university boards in future cases involve their own stakeholders in these policy planning processes. Lastly, the personnel recruited via Flycatcher and via the university mail might have been different, although they had highly comparable outcomes. That was somewhat unexpected, as the panel is based on people’s interest in university issues, while the response of all personnel might be based on interest in COVID-19, and it may indicate that the outcomes could be better generalizable than expected based on the response rates.

## 5. Conclusions

In times of COVID-19, more than half of university personnel found the university a safe place to return to in the new academic year. Still, some personnel feel unsafe for various reasons. University personnel found meeting in large groups unsettling and expressed concerns about becoming infected. In light of these worries, a prompt transition to on-site work could jeopardize their physical and psychological well-being as personnel have claimed that they require a transition period while returning to campus. These findings did assist the UM board in its decision-making process. This study demonstrates that doing research among personnel to develop adequate measures to promote employees’ safety, and their feelings of safety, was useful for university boards and may be applied in comparable future situations.

## Figures and Tables

**Table 1 vaccines-10-00371-t001:** Demographic characteristics of the participants (*N* = 1965) and comparison with population (UM).

Age	Participants	UM	Full Time?	Participants	UM
16–25	3.8%	5.1%	Yes	63.6%	60.0%
26–35	25.9%	34.1%	No	36.4%	40.0%
36–45	25.8%	23.8%			
46–55	23.2%	18.6%			
56–65	20.2%	17.7%			
>65	1.3%	0.8%			
**Gender**			**Work in**		
Female	62.2%	56.6%	A faculty	74.3%	80.3%
Male	35.2%	43.4%	A service center	22.3%	14.3%
Other	0.3%		Other	3.4%	5.3%
I do not want to answer	2.3%				
**Working at UM for**			**Being a member of a high-risk group**		
<2 years	15%	22.9%	Yes	14.4%	
2–5 years	21.9%	29.1%	No	78.4%	
6–10 years	13.9%	12.8%	I do not know	7.3%	
>10 years	49.2%	53.3%			
**Function**					
Teaching and research	45.9%	56.7%			
Academic support, policy and management	50.6%	43.3%			
Other	3.6%				

**Table 2 vaccines-10-00371-t002:** Beliefs (returning to campus in September, trust and worries, and the precautions to prevent the spread of COVID-19), mean scores, and the correlation with “When I try to imagine the situation in September, I think that UM is a safe place to return to” (*N* = 1965).

Fully Disagree (1)–Fully Agree (7)	Total Group *M* (*SD*) (*N* = 1965)	Total Group *r* (*N* = 1965)	*F*	Unsafe (1–3) *M* *(N* = 472)	Neutral (4) *M* *(N* = 349)	Safe (5–7) *M* *(N* = 1144)
** *Attitude: negative beliefs* **						
Starting again in September full-on is too fast.	3.54 (1.97)	−0.62 **	429.39 **	5.23 ^a^	4.10 ^b^	2.68 ^c^
Starting again in September full-on requires a transition period.	4.96 (1.96)	−0.54 **	342.42 **	6.25 ^a^	5.70 ^b^	4.20 ^c^
Starting again in September full-on is too risky.	3.45 (1.80)	−0.73 **	646.86 **	5.25 ^a^	4.05 ^b^	2.52 ^c^
Starting again in September full-on means that I have to protect myself against others.	4.21 (1.97)	−0.61 **	484.67 **	5.88 ^a^	4.77 ^b^	3.36 ^c^
I am afraid there will be too many adjustments for me when we return on campus in September.	3.81 (2.01)	−0.46 **	198.03 **	4.96 ^a^	4.51 ^b^	3.12 ^c^
** *Attitude: positive beliefs* **						
Returning to normal in September is certainly possible.	4.03 (1.89)	0.66 **	522.50 **	2.40 ^a^	3.36 ^b^	4.91 ^c^
I am happy that I can start working at the office again.	5.12 (1.71)	0.61 **	380.81 **	3.71 ^a^	4.56 ^b^	5.88 ^c^
I am happy that I can see my colleagues in real life again.	5.87 (1.40)	0.49 **	175.36 **	4.93 ^a^	5.58 ^b^	6.35 ^c^
I am happy that I have contact with students in real life again. †	5.45 (1.63)	0.54 **	166.82 **	4.24 ^a^	5.23 ^b^	6.08 ^c^
I think I can deal with being back in the office again.	5.37 (1.60)	0.65 **	421.42 **	3.90 ^a^	4.92 ^b^	6.11 ^c^
** *Trust* **						
I trust that UM will be a safe place in terms of people sticking to the prevention rules.	4.78 (1.76)	0.65 **	437.35 **	3.09 ^a^	4.47 ^b^	5.57 ^c^
I trust that UM will be a safe place in terms of facilities (ventilation, disinfectants).	5.20 (1.67)	0.60 **	310.95 **	3.70 ^a^	5.00 ^b^	5.88 ^c^
** *Worries* **						
I am worried about students and staff returning from high-risk countries.	4.81 (1.87)	−0.52 **	271.90 **	6.04 ^a^	5.39 ^b^	4.13 ^c^
I am worried about how to deal with vaccine deniers/refusers.	4.72 (1.96)	−0.37 **	110.36 **	5.62 ^a^	5.16 ^b^	4.21 ^c^
I am worried about meeting in large groups.	4.88 (1.89)	−0.59 **	429.95 **	6.30 ^a^	5.63 ^b^	4.07 ^c^
I am worried about my roommate(s) not being as careful as I am. ††	3.48 (2.09)	−0.53 **	192.85 **	4.99 ^a^	4.01 ^b^	2.73^c^
I think that there is too much pressure on us to be “on-site” all the time.	4.16 (2.11)	−0.59 **	390.37 **	5.81 ^a^	4.81 ^b^	3.29 ^c^
The situation with COVID-19 has improved, but I still worry about being infected.	4.28 (1.96)	−0.62 **	478.92 **	5.85 ^a^	5.11 ^b^	3.38 ^c^
** *Facilities* **						
I am sure that the ventilation at work is good enough to prevent becoming infected.	3.45 (1.71)	0.49 **	238.68 **	2.32 ^a^	3.07 ^b^	4.03 ^c^
The rules about ventilation in our buildings are not clear.	4.53 (1.71)	−0.32**	80.31 **	5.28 ^a^	4.73^b^	4.16^c^
** *Preventive measures: Entrance testing proof* **						
I think that asking people to show entrance testing proof, or to do a test, is not feasible at all–very feasible	3.43 (1.99)	−0.007	0.02	3.42 ^a^	3.45 ^a^	3.43 ^a^
I think that asking people to show entrance testing proof, or to do a test, is not useful at all–very useful	4.26 (1.94)	−0.16 **	15.14 **	4.65 ^a^	4.40 ^a^	4.07 ^b^
** *Safety guidelines* **						
In my view, the “keeping 1.5m distance” guideline should still be implemented in September, for safety.	4.98 (1.82)	−0.52 **	302.37 **	6.22 ^a^	5.52 ^b^	4.31 ^c^
In my view, the “wearing a face mask” guideline should still be implemented in September, for safety.	4.35 (2.05)	−0.50 **	233.87 **	5.73 ^a^	4.80 ^b^	3.65 ^c^
** *Dealing with health complaints* **						
People who are sniffling or coughing should stay at home and not visit the campus.	5.95 (1.45)	−0.29 **	79.07 **	6.50 ^a^	6.12 ^b^	5.67 ^c^
People who are sniffling or coughing on campus should be sent home.	5.52 (1.71)	−0.33 **	101.41 **	6.26 ^a^	5.76 ^b^	5.14 ^c^
UM should provide clear guidelines about how we should deal with students who have health complaints.	6.17 (1.12)	−0.24 **	54.12 **	6.54 ^a^	6.31 ^b^	5.98 ^c^

† A total of 30.6% of participants responded “not applicable” as they have no contact with students. When calculating the frequencies, “not applicable” (8) cases were not included in the analysis. †† A total of 18.4% of participants responded “not applicable”. When calculating the frequencies, “not applicable” (8) cases were not included in the analysis. ** *p* < 0.001;. The statistically significant mean differences were indicated with letters (i.e., a, b, and c). Each letter was used only once to show the significant difference in the means. The significance level is *p* < 0.05.

**Table 3 vaccines-10-00371-t003:** Beliefs about vaccination and mean scores of the total group, “yes” (vaccinated or intend to), “no” (decided not to get vaccinated or do not intend to), and “do not know” groups (*N* = 1965).

Fully Disagree (1)–Fully Agree (7)	Total Group *M* (*SD*) (*N* = 1965)	*F*	Yes *M* (*N* = 1860, 94.7%)	Do Not Know *M* (*N* = 66, 3.4%)	No *M* (*N* = 39, 2.0%)
*Risk perception:* Without vaccination, I might be at risk of contracting COVID-19.	**5.94 (1.48)**	90.75 *	6.08 ^a^	3.73 ^b^	3.15 ^b^
*Attitude:* Being vaccinated against COVID-19 is the only way out of this epidemic.	**5.91 (1.51)**	295.39 *	6.11 ^a^	2.77 ^b^	1.82 ^c^
*Attitude:* Being vaccinated against COVID-19 gives a feeling of safety.	**5.84 (1.44)**	240.30 *	6.02 ^a^	3.03 ^b^	1.79 ^c^
*Attitude:* Being vaccinated against COVID-19 leads to fewer negative consequences when you might be infected.	**6.18 (1.22)**	104.70 *	6.31 ^a^	4.23 ^b^	3.00 ^c^
*Attitude:* Being vaccinated against COVID-19 makes it 100% safe.	**3.09 (1.78)**	68.48 *	3.17 ^a^	2.02 ^b^	1.44 ^c^
*Attitude:* Being vaccinated against COVID-19 will lead to people keeping less distance from others.	**5.78 (1.20)**	5.14 *	5.81 ^a^	5.45 ^ab^	4.82 ^b^
*Attitude:* Vaccination is a personal and private choice; we cannot force people to take it.	**5.02 (2.03)**	125.43 *	4.92 ^a^	6.68 ^b^	6.72 ^b^
*Attitude:* We do not know how long the effect of vaccination against COVID-19 will last.	**6.10 (1.14)**	15.13 *	6.08 ^a^	6.62 ^b^	6.46 ^ab^
*Attitude:* UM should give more information about vaccination to international personnel and students.	**5.04 (1.59)**	12.77 *	5.08 ^a^	4.41 ^b^	3.82 ^b^
*Attitude:* I am worried about the safety of the COVID-19 vaccine.	**2.93 (1.80)**	436.67 *	2.75 ^a^	5.83 ^b^	6.51 ^c^
*Attitude*: I am worried about possible long-term negative side effects of the COVID-19 vaccine.	**3.12 (1.90)**	668.35 *	2.93 ^a^	6.20 ^b^	6.67 ^c^
*Attitude*: The COVID-19 vaccine will be likely effective against new mutations of the virus.	**4.14 (1.33)**	48.62 *	4.21 ^a^	3.21 ^b^	2.36 ^c^
*Attitude*: I trust the government about ensuring the safety of the COVID-19 vaccine.	**5.10 (1.58)**	102.74 *	5.23 ^a^	3.12 ^b^	2.26 ^c^
*Attitude:* By getting the COVID-19 vaccine, I can safely have more social contacts.	**5.36 (1.37)**	135.64 *	5.49 ^a^	3.48 ^b^	2.15 ^c^
*Attitude*: I think that getting the COVID-19 vaccine is my moral duty.	**5.73 (1.72)**	273.23 *	5.93 ^a^	2.20 ^b^	2.03 ^b^
*Attitude*: I would feel guilty if I transmitted the coronavirus because I had decided to not get the vaccine.	**6.14 (1.48)**	147.51 *	6.32 ^a^	3.09 ^b^	2.54 ^b^
*Social norm:* Most people like me will get the COVID-19 vaccination.	**6.03 (1.28)**	192.60 *	6.20 ^a^	3.41 ^b^	2.33 ^c^
*Social norm:* Most people who are important to me, want me to get the COVID-19 vaccination.	**5.80 (1.59)**	138.68 *	5.95 ^a^	3.21 ^b^	2.69 ^b^

* *p* < 0.01. The statistically significant mean differences were indicated with letters (i.e., a, b, and c). The significance level is *p* < 0.05.

## Data Availability

Data are available at https://osf.io/hb5rw/ (accessed on 27 January 2022).

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
