# Peer review of "A Safe Return to Campus in Times of COVID-19: A Survey Study among University Personnel to Inform Decision Makers"

_vaccines, 2022, doi:10.3390/vaccines10030371_

Round 1

Reviewer 1 Report

The manuscript is well done and brings an important overview of the COVID-19 impact on the returns of "normal" activities at UM. The study was conducted in only one center and this is the main limitation. Besides that, the is well done and written, and its findings can help other institutions regarding the same issue. The manuscript deserves to be published faster than possible.

Also, the authors could include the supplementary material as a PDF document in the submission system.

Author Response

The manuscript is well done and brings an important overview of the COVID-19 impact on the returns of "normal" activities at UM. The study was conducted in only one center and this is the main limitation. Besides that, the is well done and written, and its findings can help other institutions regarding the same issue. The manuscript deserves to be published faster than possible.

Also, the authors could include the supplementary material as a PDF document in the submission system.

Thank you for your positive feedback.

Although we wanted to wait to make the additional files public until the manuscript was accepted for publication, we now made all data publicly available under the following link:

https://osf.io/hb5rw/

Reviewer 2 Report

Thank you for the opportunity to review this manuscript. The study aimed to examine the views of university personnel as they prepare to return to work on campus. The study findings informed the university policies and it was interesting to see what was provided to the board and what was implemented. My only comments are of a minor nature.

Introduction

Page 1 line 44: “at the time being” change to the date that the vaccination program began.

Page 2 line 50: Consider including the proportion vaccinated and daily case numbers in May 2021 to add more context.

Methods

Page 2 line 78: perhaps academic personnel is better than scientific personnel

Page 3 line 130: include that was a likert scale

Page 3 line 133: Please define entrance testing here or in results, as this is not a common term

Page 3 line 147: “items can be found in supplementary material File 1”is not needed here.

Page 4 line 160: Include more about how items with Likert scale were scored – frequencies (agree and fully agree combined etc) and mean score.

Results

Page 4 line 180: Include how many personnel were sent the survey to get overall consent rate.

Page 5 line 191: “Please note…” not needed here in results

Page 5 line 204; “All items were significantly correlated” should go with the next paragraph.

Page 5 paragraph 2 and 3: refer to Table 2

Page 5 line 209: I don’t think that you mean “pregnant” – biggest perhaps? (again later as well)

Page 5 line 209: To make this more readable consider presenting results with statistics in (). Eg “The largest differences about starting in September were it being “too risky”(r=-0.73, F=429.39, p value=XXX) and …” make change throughout section.

Discussion

Page 9 line 319: For clarity consider rewording long sentence.

Author Response

Thank you for the opportunity to review this manuscript. The study aimed to examine the views of university personnel as they prepare to return to work on campus. The study findings informed the university policies and it was interesting to see what was provided to the board and what was implemented. My only comments are of a minor nature.

Thank you for your helpful suggestions and positive feedback.

Page 1 line 44: “at the time being” change to the date that the vaccination program began.

We now added “(November 2020)”

Page 2 line 50: Consider including the proportion vaccinated and daily case numbers in May 2021 to add more context.

We now added “(in the week of 11 May 2021, 84% of all people over the age of 16 vaccinated or willing to vaccinate soon)”

Page 2 line 78: perhaps academic personnel is better than scientific personnel

We changed this as suggested by the reviewer.

Page 3 line 130: include that was a likert scale

We added this as suggested by the reviewer.

Page 3 line 133: Please define entrance testing here or in results, as this is not a common term

Thank you. we now added an explanatory note to this paragraph: “(note: for entrance to restaurants, events, or other activities/buildings, a mandatory test was suggested, where people had to show that they are vaccinated, recovered from a recent COVID-19 infection, or tested negative on the coronavirus).”

Page 3 line 147: “items can be found in supplementary material File 1”is not needed here.

Our apologies. We deleted this here.

Page 4 line 160: Include more about how items with Likert scale were scored – frequencies (agree and fully agree combined etc) and mean score.

For the interested reader, all data and syntax is publicly available via https://osf.io/hb5rw/

In text, we state: “For all items, descriptive analysis was conducted to calculate the means (M), standard deviations (SD), and frequencies by using IBM SPSS Statistics 26”

Page 4 line 180: Include how many personnel were sent the survey to get overall consent rate.

We added this number on p2, line 79 (under ‘procedure and participants).

On page 4, we added: “Of the 7198 invited people, a total of 1965 personnel (27.3% response rate)”.

Page 5 line 191: “Please note…” not needed here in results

We deleted this, as suggested by the reviewer.

Page 5 line 204; “All items were significantly correlated” should go with the next paragraph.

We changed this as suggested by the reviewer.

Page 5 paragraph 2 and 3: refer to Table 2

We now referred to Table 2 in both paragraphs.

Page 5 line 209: I don’t think that you mean “pregnant” – biggest perhaps? (again later as well)

We changed this to “the largest correlations with…” (3x)

Page 5 line 209: To make this more readable consider presenting results with statistics in (). Eg “The largest differences about starting in September were it being “too risky”(r=-0.73, F=429.39, p value=XXX) and …” make change throughout section.

Throughout the section, we now put the r-values between brackets to improve readability. We did not add all other statistics (F and p), as they can be found back in Table 2 (and it would reduce the readability again).

Page 9 line 319: For clarity consider rewording long sentence.

We changed this to: “In the Netherlands, all university personnel and students can get vaccinated against COVID-19. In this study, we found that most university personnel were either already vaccinated or intended to get vaccinated (94.7%).”

Reviewer 3 Report

Varol et al. performed a survey among university staff with the aim of deepening the feeling of safety in view of the reopening of the Campus in the Netherlands.This is an interesting study on a relevant public health topic that provides useful information to ensure the safe resumption of work activities in a pandemic era. Methods appear to be sound, data are nicely presented and information provided adds knowledge to this topic area and may be of stimulus to other researchers in the investigation in this and other areas.

Some minor essential revisions are following:

Page 4, line 180. Authors reported a total sample of 1965 subjects included in the study but it is not clear what was the response rate. Please, clarify.

Table 1. In the first strata of the age categories subjects aged 16-17 years were included. Are these participants classified as University personnel? And if so, in which role? 

In relation to the function of the University personnel what was included in the “other” category? Please add the description in the methods section (demographics paragraph, line 151).

In the Discussion section, it would be interesting to compare the principal results of the study with those obtained in previous similar surveys conducted in other countries.

Author Response

Varol et al. performed a survey among university staff with the aim of deepening the feeling of safety in view of the reopening of the Campus in the Netherlands.This is an interesting study on a relevant public health topic that provides useful information to ensure the safe resumption of work activities in a pandemic era. Methods appear to be sound, data are nicely presented and information provided adds knowledge to this topic area and may be of stimulus to other researchers in the investigation in this and other areas.

Thank you for your positive feedback and suggestions

Page 4, line 180. Authors reported a total sample of 1965 subjects included in the study but it is not clear what was the response rate. Please, clarify.

Thank you. please see reviewer 2, #10: We added the total number of invited participants on p2, line 79 (under ‘procedure and participants).

On page 4, we added: “Of the 7198 invited people, a total of 1965 personnel (27.3% response rate)”.

Table 1. In the first strata of the age categories subjects aged 16-17 years were included. Are these participants classified as University personnel? And if so, in which role? 

We received these general age-groups (in this case 16-25 years) from the panel we worked with. We assume there were no 16/17 year olds in this group, but we don’t have additional data available.

In relation to the function of the University personnel what was included in the “other” category? Please add the description in the methods section (demographics paragraph, line 151).

This rest-category was very small (3.6%): we were mainly interested in the two larger categories (1) Teaching & Research, and (2) Support, Policy, Management.  We do not have data on what “other” functions. We now included “”Other” [not further specified]”

In the Discussion section, it would be interesting to compare the principal results of the study with those obtained in previous similar surveys conducted in other countries.

We were unable to find comparable studies. Therefore we added in the discussion: To our knowledge, there are no comparable studies published yet in this (or similar) setting and/or context.